

# Are dependencies of extreme rainfall on humidity more reliable in convection-permitting climate models?

Geert Lenderink[1], Nikolina Ban[2], Erwan Brisson[3], Ségolène Berthou[4], Virginia Edith Cortés-Hernández[3], Elizabeth Kendon[4,5], Hayley J. Fowler[6], Hylke de Vries[1]

[1]R&D Weather and Climate, Royal Netherlands Meteorological Institute (KNMI), De Bilt, The Netherlands
[2]Department of Atmospheric and Cryospheric Sciences, University of Innsbruck, Austria
[3]Centre National de Recherches Météorologiques (CNRM), Université de Toulouse, Météo-France, CNRS, Toulouse, France
[4]Met Office Hadley Centre, Exeter, UK
[5]Bristol University, Bristol, UK
[6]School of Engineering, Newcastle University, Newcastle upon Tyne, UK

*Correspondence to*: Geert Lenderink (geert.lenderink@knmi.nl)

**Abstract.** Convection-permitting climate models (CPMs) are becoming increasingly used in climate change studies. These models show greatly improved convective rainfall statistics compared to parameterized-convection regional climate models (RCMs), but are they also more reliable in a climate change setting? Increases of rainfall extremes are generally considered
to be caused by increases in *absolute* humidity, primarily following from the Clausius-Clapeyron relation, while the influence of *relative* humidity changes is uncertain and not systematically explored. Quantifying these humidity dependencies in the present-day climate may help the interpretation of future changes, which are driven by increases in absolute humidity, but also decreases in relative humidity in most continental areas in summer. Here, we systematically analyse hourly rainfall extremes and their dependencies on 2m dew point temperature (absolute humidity) and dew point depression (relative humidity) in 7
RCM and 5 CPM simulations for the present-day climate. We compare these to observations from The Netherlands (a moderate moist climate) and Southern France (a warmer and drier climate). We find that the RCMs display a large spread in outcomes, in particular in their relative humidity dependence, with a strong suppression of hourly rainfall extremes in low relative humidity conditions. CPMs produce better overall rainfall statistics, show less inter-model spread, and have absolute and relative humidity dependencies more consistent with the observations. In summary, our results provide evidence that future
changes in convective rainfall extremes in CPMs are more reliable compared to RCMs, whereas the discussed dependencies also provide a metric to evaluate and further improve model performance as well as improving convection schemes.

## 1 Introduction

In recent years, climate modelling has rapidly progressed towards higher resolution with so-called *convection-permitting* climate models that explicitly resolve the largest convective motions in the atmosphere (Stevens et al., 2019; Prein et al., 2020;
Schär et al., 2020; Ban et al., 2021; Kendon et al., 2021b; Pichelli et al., 2021; Lucas-Picher et al., 2021). These models typically run at <4-5 km grid spacing, with non-hydrostatic dynamics, which allows them to explicitly resolve deep convection





without the need for a convection parameterization. As computational demands are high, CPMs can only be applied for relatively short time periods; on regional domains typically 10-30 years are simulated. In recent years, large efforts have been made to produce a number of these CPM runs, for instance the international coordinated action CORDEX flagship pilot study

over the European Alpine region (Coppola et al., 2020; Ban et al., 2021; Pichelli et al., 2021) and a 12-member ensemble for 100 year projections from 1980 to 2080 produced as part of the UK national climate scenarios (Kendon et al., 2020, 2021a, 2023).

The primary reason to run CPMs at km-scales is their improved representation of convective processes, which give rise to

locally-intense convective rainfall on sub-daily time scales that e.g. can lead to flash floods and debris flows. In recent years, knowledge on these type of extremes, and how they could be affected by climate change, has advanced considerably based on observational evidence, process understanding, and high-resolution modelling such as with CPMs (Fowler et al., 2021a, b). Convection parameterization schemes have been identified as responsible for substantial biases in convective rainfall statistics in coarser-resolution climate models, causing rainfall intensities that are too low and too frequent, and a too early triggering

during the day (Kendon et al., 2012; Ban et al., 2021). CPMs improve the frequency and diurnal cycle of rainfall, as well as the distribution of the extremes and rainfall spatial patterns (Kendon et al., 2012; Ban et al., 2020, 2021; Berthou et al., 2020; Fumière et al., 2020). CPMs have also been evaluated for process-based metrics such as intensity-duration characteristics and cell-size distribution (Kendon et al., 2012), assessing the reliability of the underlying rainfall processes. Finally, rain cell tracking has been used to investigate the lifecycle of convective rainfall events, providing additional information on how

organized convective systems behave in CPMs versus radar data (Purr et al., 2019, 2021; Müller et al., 2023).

Limited evidence exists that the improved representation of convective cloud processes in CPMs, as compared to RCMs, leads to more certain estimates of future extreme rainfall changes (Fosser et al., 2020, 2024). Indeed, when future changes are dominated by simple thermodynamic factors (the moisture increase) then CPMs often project similar changes to RCMs (Ban

et al., 2020). Despite their improved behaviour, CPMs are not without faults: at 2-3 km resolution important parts of the physics and dynamics of convective clouds are still not (well) resolved, e.g. convective plumes, entrainment processes at the cloud interface, detailed boundary layer cold pool dynamics and cloud microphysical processes (Prein et al., 2021; Lochbihler et al., 2021). Therefore, it is important to establish the trustworthiness of these models using measures that relate to climate change. Here, we study the dependency of extreme rainfall on measures of absolute and relative humidity – key drivers of

future changes, as explained below.

The intensification of rainfall extremes due to global warming is to first order explained by increases in absolute humidity following from the Clausius-Clapeyron (CC) relation (Pall et al., 2006; Trenberth, 2011; Fischer and Knutti, 2016). The Clausius-Clapeyron relation governs the increase in the water vapor *holding* capacity of the air as a function of temperature,

and near to the surface gives a rate of increase of 6-7% per degree warming. Evidence from observations suggests that the rate





of change of precipitation extremes with absolute humidity could be much stronger. In observations, so-called scaling rates of precipitation extremes on near surface point temperatures up to 12-14% per degree (2xCC) have been found (Lenderink and van Meijgaard, 2008, 2010; Lenderink et al., 2011; Fowler et al., 2021a; Ali et al., 2022) as well as strong rainfall intensification beyond a critical value of the integrated water vapor path (Neelin et al., 2022). Scaling rates beyond the CC relationship are

commonly denoted as super-CC scaling. But scaling rates also differ for different areas of the globe, and depend on the temperature measure used (Lenderink et al., 2018; Bui et al., 2019). It is not always straightforward to connect the temperature measure used to the humidity of the air in which a rain shower develops. In this paper, we use dew point temperature which directly measures absolute humidity.

Several explanations have been proposed to explain observed super-CC scaling. These range from systematic statistical shift in the rainfall type with temperature (Haerter and Berg, 2009; Fowler et al., 2021a; Molnar et al., 2015), to positive feedback mechanisms from physical processes. Latent heat release in the cloud could invigorate updraft motions, leading to more condensation of moist updraft air, and stronger precipitation rates (Loriaux et al., 2013). Stronger sub-cloud cold pool dynamics with warming could lead to stronger cloud organisation and larger moisture availability at the cloud condensation level (Haerter

and Schlemmer, 2018; Lochbihler et al., 2021). Deeper warm cloud levels below the freezing level could lead to more efficient warm rain processes (Prein and Heymsfield, 2020). Finally, large-scale dynamical adjustments to latent heating could also lead to enhanced moisture convergence and destabilization of the atmosphere (Lenderink et al., 2017; Nie et al., 2018).

We do not aim here to further study the causes of super-CC behaviour. Instead, we conjecture that reproducing the observed

near-surface dew point temperature dependence in a model is an essential prerequisite for confidence in its future projections. Nonetheless, we acknowledge that the relation between present-day derived scaling rates and future projection of changes in rainfall extremes is rather complex (Lenderink and Attema, 2015; Bao et al., 2017; Zhang et al., 2017; Lenderink et al., 2019, 2021; Fowler et al., 2021a). One major complication is that co-varying atmospheric variables with (dew point) temperature – such as for example, large-scale circulation or atmospheric stability – may have a different correlation in day-to-day variability

as compared to long time climate change (Lenderink et al., 2017; Fowler et al., 2021a). Thus, although it is important to reproduce observed scaling rates, it is not a sufficient requirement.

In contrast to the anticipated increase of rainfall extremes due to absolute humidity change, the sign of the extreme rainfall response due to relative humidity change is very uncertain (Fowler et al., 2021a, b). This is important as the summer season,

in which most of the convective extremes occur in mid-latitudes, is expected to see substantial reductions in relative humidity in a warmer future climate (Lenderink and Attema, 2015; Byrne and O'Gorman, 2018; Williams and O'Gorman, 2022; Zhou et al., 2023). Links to relative humidity can be very complex. Reductions of relative humidity may lead to a smaller number of showers (Dai et al., 2020), for instance because (near surface) air needs to rise to higher levels to reach saturation reducing the cases where convection is triggered. In addition, low relative humidity in the cloud layer leads to a strong dilution of





convective updrafts by entrainment processes (Derbyshire et al., 2004), possibly leading to weaker convection and reductions in rainfall extremes. However, lower RH is also connected to larger values of convective inhibition, leading to a longer build-up of convective instability and therefore stronger updraft motions (Rasmussen et al., 2020). Enhanced warming of the surface layer before convection is triggered could compensate the stabilization (in the dry lapse rate) of the atmosphere due to enhanced warming of the upper air; this stabilization occurs as a consequence of a (partly) moist adiabatic adjustment of the atmosphere

(Loriaux et al., 2013; Attema et al., 2014). In addition, the stronger evaporation of rain in a drier and deeper atmospheric boundary layer may lead to stronger cold pool dynamics, promoting larger and more organized convective cloud systems (Lochbihler et al., 2021). Finally, deeper and drier boundary layers could lead to wider updrafts at the cloud base, eventually leading to stronger convection (Mulholland et al., 2021). It is therefore important that the dependency of precipitation extremes on relative humidity is well understood and modelled.


Within the context of absolute and relative humidity changes, this paper aims to answer the following questions:

- What is the relationship between absolute and relative humidity and precipitation extremes? And how can we establish these relationships with simple measures?
- How do these relationships in parameterized-convection and convection-permitting models compare to observations?
Are convection-permitting models substantially better?
- How can we use this information to enhance our (confidence in) future projections of extreme rainfall?

We use data from The Netherlands – a relatively mild and humid climate – and the southern part of France – warmer with lower relative humidity – to study these questions. We also specifically examine commonalities and differences between these

two regions. This is potentially interesting since the future climate of the Netherlands is projected to be more like the present-day climate of southern France, with higher temperatures, lower relative humidity and likely decreases in rainfall frequency (Lenderink et al., 2014; Aalbers et al., 2022). We perform a series of analyses to answer the questions above, using scaling on dew point temperature and then further stratifying the data on dew point temperature depression. To complement these analyses, we also examine the distributions of absolute and relative humidity conditional on different rainfall intensity

classifications (Lenderink and van Meijgaard, 2010; Lenderink et al., 2011).

## 2 Methods

### 2.1 Observations and models

Two observational data sets are used, both with hourly accumulated rainfall and hourly temperature and humidity measurements. For the Netherlands (NL), we use 33 automatic weather stations distributed rather evenly over the country and

operated by the Royal Netherlands Meteorological Institute from 1991 to 2020. Most stations have almost complete time series (25 stations with >90% data). For southern France (SFR), 59 stations are used that have at least 10 years data coverage





in the period 1991-2020. These stations are also selected based on their altitude below 400 meters to limit orographic effects. In earlier work we found that scaling relations of hourly extremes derived from these lower altitude stations are quite robust, and not dependent on either the time period or region analysed within western Europe (Lenderink et al., 2021), which is
reaffirmed here by considering the differences between central France and The Netherlands. On average the stations in SFR contain 16 years of data, with a maximum of 25 years. Since we pool station data together (and do not examine stations individually) the dataset is large, containing over 900 years of data.

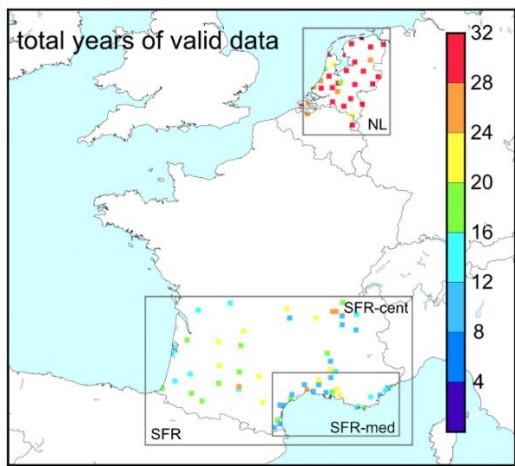

**Figure 1: Map of the stations used in the study. The colour scale indicates the number of total years of valid data for each of the stations.**

We use outputs from 7 RCMs: CLMcom, HadRM3, RACMO, RCA4, HIRHAM5, REMO and ALADIN. References for these models can be found in the Supplementary Information. All models have been run at 12 km grid spacing, and we analyse the
period from 1991-2010. All models use a convection parameterization, both for deep as well as shallow clouds.

We use 4 different CPMs: HCLIM, AROME, COSMO and UKMO-UM. These simulations are performed on a domain extending sufficiently north to cover The Netherlands. Typically, the runs are 10-12 years, with at least the period 2000-2009 covered in all models (see Supplementary Information for details). For HCLIM two runs are analysed: HCLIM-ALP covers a
large domain around the Alps (as prescribed in the CORDEX-FPS study) for the period 1999-2010, and HCLIM-NWE covers a large domain in north-west Europe for the period 2008-2018. All models run without a deep convection parameterization, but some keep a convection parameterization for shallow convective clouds.





All model runs are driven by ERA-Interim reanalysis data (Dee et al., 2011), providing realistic atmospheric forcing boundary
conditions. For the CPMs, UKMO is directly nested into ERA-Interim, but the other models use an RCM as an intermediate
nest, providing boundary conditions at high spatial and temporal resolution.

## 2.2 Analysis

All analysis uses data pooled over many stations (or grid points for model data). Data is pooled for the Netherlands over all
stations (NL). For southern France (SFR) we separate the data set into 25 stations close to the Mediterranean coast (lat <
43.8°N, lon > 3.8°E, labelled SFR-med), and 34 stations more inland, labelled SFR-cent (Figure 1).

We mainly show results for the summer period from May to September (MJJAS) in which most of the convective extremes
occur in the Netherlands and SFR-cent. But we also briefly look at the spring to mid-summer period, April to July (AMJJ) and
the late-summer to autumn period, August to November (ASON). Results for SFR-cent and NL are very robust, with very
distinct scaling behaviour only (very) weakly dependent on the season analysed. In contrast, results from SFR-med reveal how
a more complex meteorology, with influences from the Mediterranean Sea as well as the nearby orography, affects the results.

For the CPMs, we first re-grid the hourly data based on blocks of 5x5 grid points. For these 5x5 grid blocks, we compute the
mean (Pmean$_{5x5}$) and take the value from the point centred in the middle of the 5x5 box (Psample$_{5x5}$, called *sample* as it samples
the 25 grid points in the block). By comparing Pmean$_{5x5}$ with Psample$_{5x5}$ we can establish the dependency on spatial scale, and
the scale of Pmean$_{5x5}$ is approximately comparable to RCM grid box scale. Here, we use Psample$_{5x5}$, which is most
representative for the local station observation. However, we note that very similar results are obtained using Pmean$_{5x5}$, with
dependencies not strongly dependent on spatial scale.

The closest grid point to the station location is used, without spatial interpolation. For the model results we take the complete
time series, and we do not correct for missing observational data by neglecting the same hours from the model results.

Uncertainties in the observations are computed using a bootstrap procedure. We take 10 random years from the 1991-2020
period and calculate all analysis statistics on the 10 year sampled data. We note that we take 10 years as comparison to CPM
model simulations with a ~10-year length. We take 100 bootstrap samples and plot the 5-95% range of samples. Since the
data in France contains more missing data, this implies that uncertainties in the observations for this area are somewhat larger
than in the models, yet experimenting with 15-year bootstrap samples only gave small differences.

In this study we use the atmospheric temperature T, dew point temperature TD, and dew point depression, DPD = T – TD, all
near the surface at 2m. The dew point is defined as the temperature at which an air parcel reaches saturation (100% relative
humidity) when cooled adiabatically and at constant pressure; TD is therefore a measure of absolute humidity, and each degree





rise in TD reflects an increase of 6-7% in the humidity of the air following the CC relation. The dew depression is a good measure of *relative* humidity, which follows from the fact that the saturation specific humidity varies approximately exponentially with temperature (according to the CC relation). Each degree of DPD increase represents an approximately 4-

6% drop in relative humidity.

Dependencies on TD (absolute humidity) and DPD (relative humidity) are computed from two procedures (following Lenderink and van Meijgaard 2010). For both procedures we first pair the instantaneous values of dew point temperature to the precipitation 4 hours later (or closest to that). We note that some models only provide 3-hourly (dew point) temperature;

hence, using a temperature 4-hours prior guarantees that the (dew point) temperature is taken before the precipitation, so that it is not likely strongly affected by the rainfall event itself. Using a lag of 4 hours also provided the most robust results in earlier scaling analyses based on observations and, in physical terms, limits the influence of showers themselves on the surface temperature by cold (and dry) downdrafts (Lenderink *et al* 2011). All further analysis is done on this paired data set (temperature on a wet hour paired with temperature at 4 hours preceding the hour with rainfall).


We first compute TD and DPD statistics conditional on rainfall intensity. We sub-select pairs based on rainfall intensity, taking all hours including those without rain (all), all hours with rain exceeding 0.1 mm hour$^{-1}$ (wet), and hours with rain exceeding the 50, 90, 95, 99, 99.5 and 99.9$^{th}$ percentiles of rain (with the percentile computed from wet hours only). Subsequently, we compute various statistics from the selection of TD and DPD, for instance, the median TD.


For the scaling analysis, we compute rainfall intensity conditional on TD. First, the pooled data is divided into two-degree wide TD bins. As in previous studies we use overlapping bins with steps of one degree. Conditional percentiles (90, 99, 99.9$^{th}$) are computed for each bin, taking only hours with rainfall exceeding a threshold of 0.1 mm hour$^{-1}$. Besides binning on TD only, we also distinguish between different relative humidity classes based on DPD, with DPD < 3 °C marking high relative

humidity, DPD from 3 to 6 °C marking medium relative humidity, and DPD > 6 °C marking low relative humidity. This procedure divides the data into three approximately equal parts, although obviously the drier climate in SFR leads to more data in the low relative humidity class.

Finally, in some figures with many lines from different models, results are slightly smoothed using a LOESS filter (Chan et

al., 2016) in order to make the figures less cluttered and easier to understand (this is mentioned in the figure caption). Bootstrap uncertainties are only given when the statistics can be derived in more than 90% of the bootstrap samples. Percentiles are only computed when there are more than 20, 200, and 2000 data points for the 90, 99 and 99.9$^{th}$ percentiles, respectively. The distributions of dew point and dew point depression based on rain intensity class are only computed when there are more than 100 paired data points. Some statistics are plotted outside the range for which the bootstrap uncertainty could be established



(note that the bootstrap uses only 10 years instead of the 30 years in the full data set), but obviously the values in this case are very uncertain.

## 3 Results

### 3.1 Extreme statisics

Before analysing temperature and humidity dependencies, we first examine the rainfall distribution derived from the pooled

dataset for both regions in southern France, SFRA-cent and SFR-med, and the Netherlands, NL. Plotted in Figure 2 is the *probability* of exceedance derived from the pooled dataset over all stations and all time-steps. The most striking result is that the ensemble of RCMs has a much larger spread than the CPM ensemble. All of the RCMs underestimate hourly rainfall for NL up to moderate intense events (10 mm hour$^{-1}$). In the extreme tail about half of the models still substantially underestimate, whereas the others catch-up; even leading to an overestimation for HIRHAM5. We note that these high extremes in the RCMs

could be related to unphysical grid-point storms (Chan et al., 2014). In RACMO, however, we also observe that the *resolved* dynamics produce a convective-system-like up and downdraft when the atmosphere is very moist, producing a convective rain systems that is too large scale and too persistent (see Fig. S8). For SFRA-cent, the situation is similar, yet one model (HadRM3) appears to be surprisingly good except for a very few high intensity events (>50 mm hour$^{-1}$).

CPMs generally show much more consistent results, with the majority of the models close to the observations for intensities below 20 mm hour$^{-1}$. However, except for UKMO-UM, they appear to underestimate the extreme tail of the rainfall distribution in SFRA-cent and SFR-med; a behaviour that is also apparent for NL, with models not exceeding 50 mm hour$^{-1}$, with the exception again of UKMO-UM which is now too extreme. Summarizing, in line with previous studies, CPM results show more consistent behaviour across the multi-model ensemble, with extreme rainfall distributions closer to the observations than

for the RCMs.





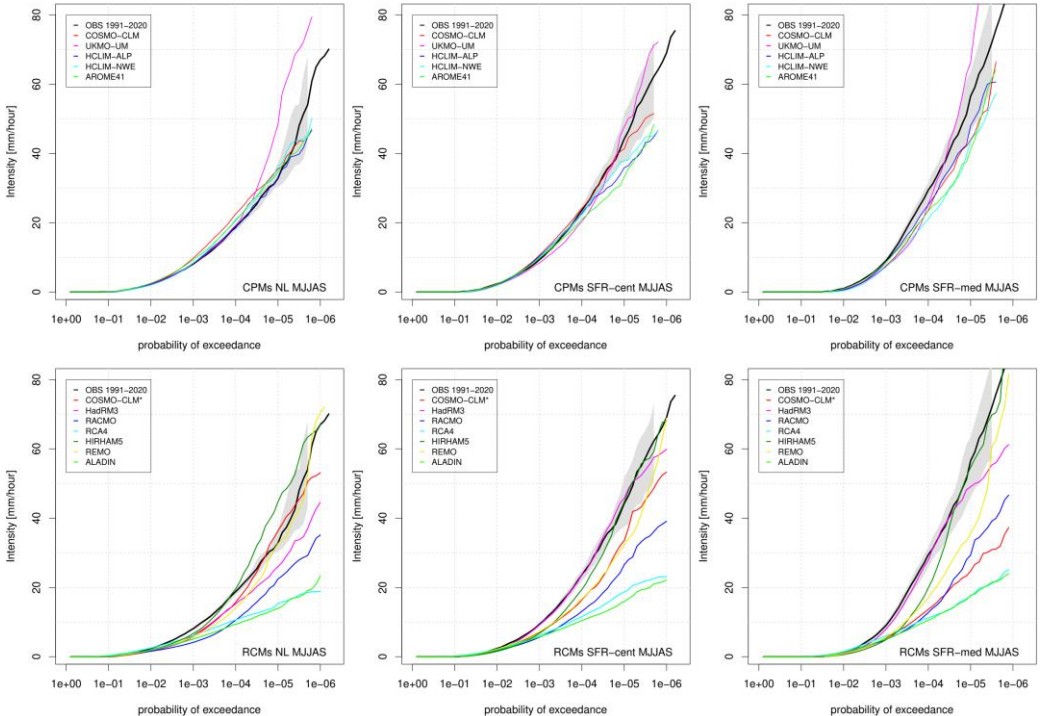

**Figure 2: Probability of exceedance derived from the pooled data of all stations for MJJAS. Grey bands around the observations for**
**1991-2020 are 90% uncertainty ranges estimated from bootstrapping.**

### 3.2 Dependencies conditional on rain intensity

Before discussing dependencies on rainfall intensity, we first look at the climatological distribution of near-surface dew point temperature, TD and DPD. Results for NL and SFR-cent are shown here in Figure 3 (left point labelled with "all" in graph). Since CPMs and RCMs are, except for higher resolution, not fundamentally different in many aspects of the parameterized processes in the soil and in clouds, we do not necessarily expect better behaviour from the CPMs here. In addition, the larger intermittency and higher rainfall intensity in CPMs may lead to more runoff and drier soils (Berthou et al., 2020). Indeed, looking at the median TD for all hours and the median TD for wet events in SFR-cent, CPMs do not show better behaviour than RCMs, with both HCLIM as well as UKMO-UM showing a dry bias corresponding to the low-end of the RCM ensemble. This dry bias is fairly constant across the full TD distribution in most, but not all, models indicating that the variability in TD is approximately correct (Fig. S2, panels on the left).

As expected, higher intensity rainfall events occur on average with higher TD, gradually increasing with the extremity of the rainfall event; see Figure 3 (left panel) showing the median TD for different rainfall intensity classes. CPMs and RCMs display rather similar behaviour in this respect. Typically, the slope of the line is parallel to the observed slope showing that a model with a dry mean bias typically also produces extreme rainfall at too low absolute humidity, and vice versa. This general behaviour is also obtained by investigating the full distributions of TD, instead of only the median shown in Figure 3 (see Fig.





S2). There appear to be no substantial systematic shifts in the anomaly of TD in the two model ensembles with rainfall intensity class (all, wet and 99th percentile). Thus, the differences in the extreme rainfall statistics (Figure 2) between the RCMs and CPMs cannot be easily explained by differences in absolute humidity alone.


The observed dewpoint depression, DPD, shows a moderate increase with rainfall intensity class for NL and a more pronounced increase for SFR-cent, revealing that more intense rain occurs on average at lower values of relative humidity (Figure 3). For SFR-med (Fig. S1) there is almost no dependency, which may be related to the more complex meteorology connected to rainfall extremes in this region (Duffourg and Ducrocq, 2013). The CPM ensemble represents this dependency

rather well. But, particularly for SFR-cent, the spread is considerable and outside of the observed uncertainty ranges, with most CPMs simulating too high DPD (too low relative humidity). The RCMs are clearly biased. Except for HadRM3, they all require too low values of DPD (too high relative humidity) to produce extreme events. This is the case for all three regions studied (Figure 3, and Fig. S1). Potential causes of this behaviour – related to how these models represent convection – will be elaborated on in the discussion.




**Figure 3: Median dew point temperature, TD, conditional on hourly rainfall intensity (left), and median (middle) and 80th percentile (right) of the dew point depression, DPD; for all hours (that is, climatology, including dry hours), wet hours, and those with hourly rainfall exceeding the conditional 50, 90, 95, 99, 99.5, and 99.9th percentiles. First two rows show results for NL, and last two rows show results for SFR-cent; results of the CPMs in 1st and 3rd row, and RCMs in 2nd and 4th row. We use paired precipitation and TD time series (TD are 4 hours prior to rainfall). (Statistics are only computed with more than 100 paired measurements, causing the shorter 10-year bootstrap samples to give no data for the 99.9th percentile rain intensity class).**





To further investigate the robustness of this result, Figure 4 shows the full distribution of the DPD anomaly compared to the observations for rainfall events exceeding the 99th percentile. The difference between the RCMs and CPMs is again obvious; the RCMs are almost all substantially below the zero line, reflecting the occurrence of rainfall extremes at predominantly too high relative humidity. The CPMs, however, do not show this behaviour, and many of them even seem to display a reversed dependency; they tend to produce heavy rainfall at too low relative humidity.


Understanding the underlying causes of the differences between the model derived dependencies and observations is difficult with such a statistical analysis. A model bias in the DPD distribution for extreme events could be related to a bias in the DPD climatology of the model but could also reflect differences in how convective rainfall responds to relative humidity. As an example, it could be that convective rainfall in a model is only triggered in case the boundary layer is close to saturation due

to convection parameterization (Hohenegger et al., 2009). In this case, a model could have mean bias with too low relative humidity, but still display too high relative humidities for extreme rainfall events.

Thus, we argue that by considering how the anomaly in dew point and dew point depression depends on rainfall intensity class we can get a hint towards the causal mechanisms. Our conjecture is that when these anomalies depend strongly on the intensity

class, the role of parameterization and convection-related mesoscale processes is important. In contrast, when it is mainly independent then other mechanisms – for example, related to soil memory and runoff processes – could be more relevant. But because a large fraction of the total rainfall falls as "extreme" and because runoff is a function of rainfall intensity these two effects are intrinsically intertwined.

Acknowledging the subtleties in the interpretation described above, the RCMs show a very clear shift to high relative humidity (low values of the dew point depression) for high intensity rainfall events (Figure 3, Figs. S1-3). This behaviour is most strongly present in SFR-cent, but also visible for NL and SFR-med (Figure 4, and Figs. S1-3). There is one remarkable exception to these results: the Met Office model (HadRM3) is almost bias free in this respect. For the CPMs, however, we do not observe such a dependence on rainfall intensity, suggesting that general model climatological biases, for instance due to

dry soils, are more important.





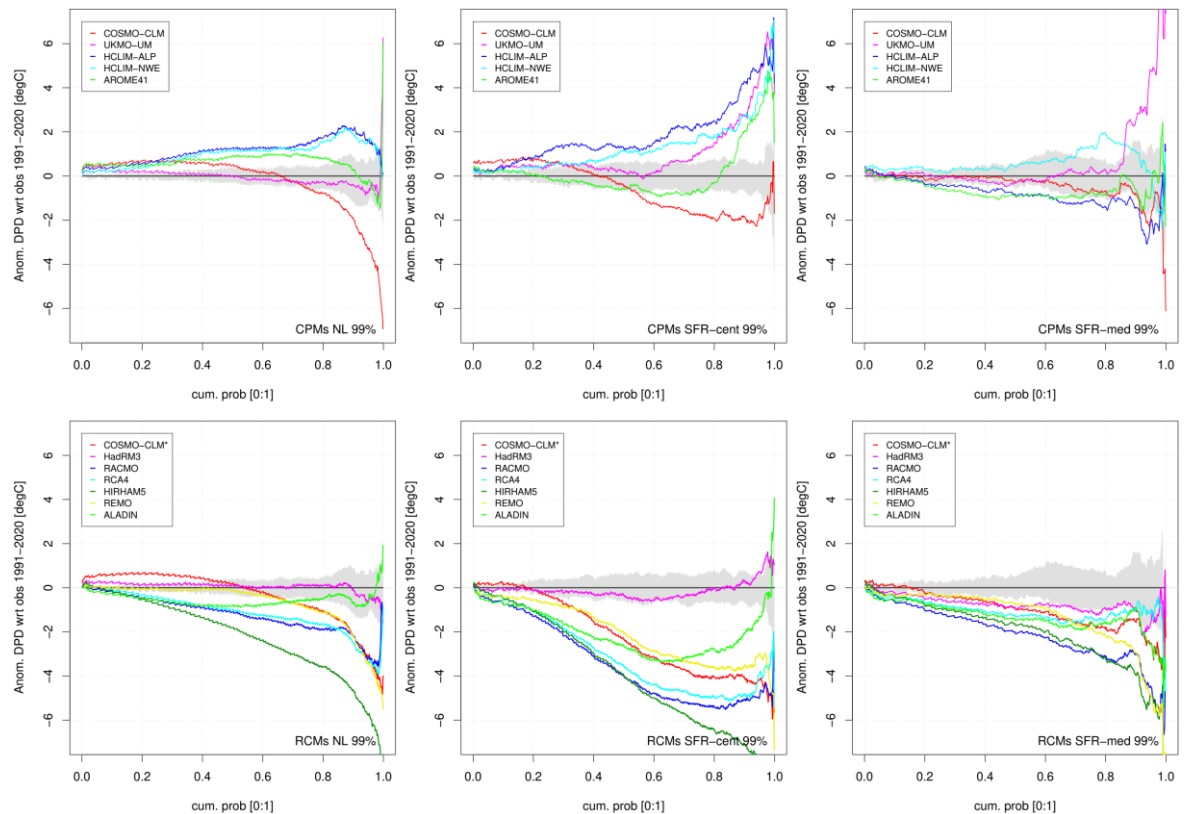

**Figure 4: Anomalies with respect to the observations (1991-2020) of the distribution of dew point depression, DPD, for events exceeding the 99th percentile of rainfall intensity. Plotted is the delta between model and observations as function of the cumulative probability, similar to quantile difference plot.**

### 3.3 Dependencies conditional on absolute and relative humidity

Humidity dependencies are studied in more detail using a scaling analysis. Figure 6 shows that the highest percentiles, the 99 and 99.9th, clearly reveal close to 2CC dependencies (13% per degree) on TD for the observations in NL and SFR-cent. This behaviour is surprisingly robust for both areas, with almost the same behaviour obtained by analysing the spring/early summer period (AMJJ) and late summer/autumn period (ASON) (Fig. S4). Following earlier studies, we use conditional percentiles – investigating rainfall intensity preconditioned on the occurrence of rain – as these show the most consistent behaviour; e.g. results show larger bootstrap uncertainty bands and less consistent scaling rates across the TD range using absolute percentiles. This is consistent with rainfall intensity rather than rainfall frequency being related to the moisture availability in the atmosphere. But we note that the results using absolute percentiles are quite similar, with only somewhat worse behaviour for the lowest percentiles (Fig. S5; and see results in the discussion).



For the SFR-med, positive dependencies on dew point temperature are obtained, but these dependencies are less robust and there are quite strongly varying scaling rates with dew point temperature (Figure 5) as well as seasonality effects (Fig. S4). The indicates local near surface humidity is not the main source of moisture (or not correlated well with it) for rainfall (e.g. moisture at higher levels may be more important) or other factors (e.g. specific dynamical circulation patterns) are important

for controlling rainfall intensity. This may reflect the more complex topography and associated meteorology of the region. Given the less reliable scaling results for SFR-med, we hereafter focus our analysis on results from NL and SFR-cent.

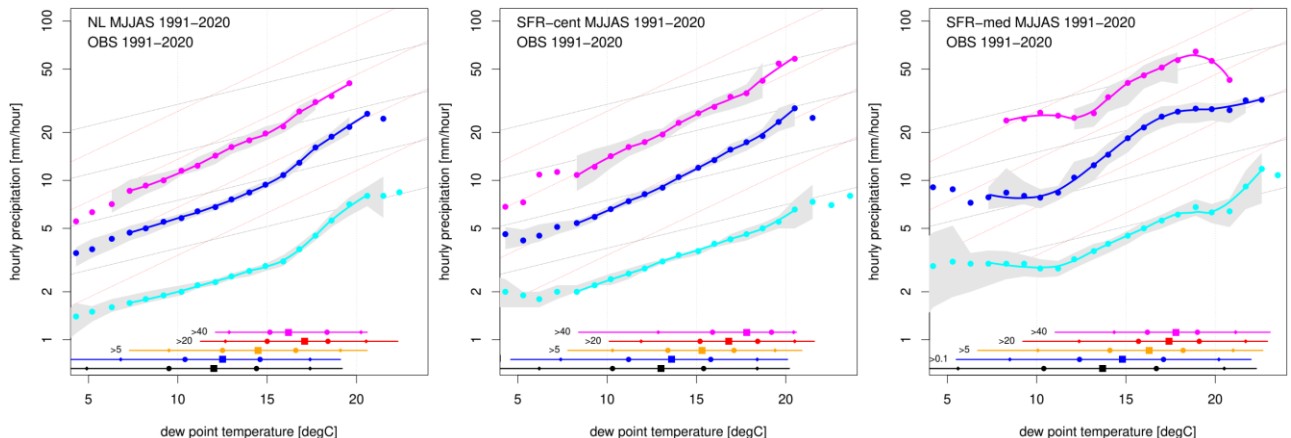


**Figure 5: Scaling of hourly observations of precipitation extremes on dew point temperature, TD, with cyan, blue, and magenta lines showing the 90, 99, and 99.9th percentiles based on wet hours only. Dotted black and red lines show a dependency of 1 and 2 times the CC relation. Horizontal lines at the bottom show the dew point temperature range for all hours (black), hours with rain (blue) and hours with rain exceeding 5 (orange), 20 (red) and 40 (magenta) mm, with the line showing the 1 to 99th percentile range and**
**markers indicating the 5, 25, 50 (square), 75, 95th percentiles. Grey bands are uncertainty estimates from the bootstrapping procedure. Loess smoothed lines are plotted for a relevant TD range (orange line, showing the 1-99th percentile range of TD for hourly rainfalls exceeding 5 mm), and uncertainty bands are plotted where more than 90% of the bootstrap samples contain sufficient data (plots for individual models are given for NL and SFR-cent in the supplementary information).**

For further investigation we use the 99th percentile, which is a compromise between analysing sufficiently extreme events and
still containing sufficient data for a robust estimate. For this percentile we further classify on DPD as a measure of relative humidity. In general, scaling rates – the rate of increase in intensity per degree TD – are rather similar for the different DPD classes. Results for SFR-cent (Figure 6) and NL (Fig. S6) all show a dependency close to the 2CC rate, with only small variations with dew point temperature.

The CPMs generally reproduce the observed dependencies rather well (Figure 6, upper panels, and Fig. S6). Most CPMs tend to overestimate the rainfall intensities for SFR-cent and NL; this is most prominent for high relative humidities. The RCMs reveal much more diversity in scaling behaviour (Figure 6, lower panels). Generally, their hourly rainfall amounts are substantially too low. But some model results appear to be reasonable; for example, HadRM3 overall shows good comparison





with observations. Yet, other RCMs show very divergent behaviour. HIRHAM5 is an (extreme) example. The model has

reasonable dew point temperature scaling for high relative humidity, but is far off for low relative humidity (consistent with the erroneous relative humidity dependency of HIRHAM5 shown in Figure 4). This inability to reproduce the correct TD scaling for low relative humidity is apparent in most RCMs; for instance, for SFR-cent most RCMs show hardly any dependency between 10 and 20 °C (RACMO, RCA, REMO, HIRHAM5). As relative humidity is expected to decrease in the future climate, this could imply that expected increases in rainfall extremes are underestimated in these models.


To summarize these findings, we plotted rainfall amounts at 15 °C against the scaling coefficient for the 99th percentile in observations (black, with estimates of uncertainty from the bootstrap), RCMs (open triangles) and CPMs (solid squares) (Figure 7). These results are derived from fitting a linear dependency to the logarithmic of precipitation between 10- and 20-degrees dew point temperature and plotting the fit coefficient against the value at 15-degrees. For the observations of NL and

SFR-cent this is rather accurate (and resulting errors are small) since the scaling lines in Figure 5 are almost linear in that dew point temperature range. For all relative humidity classes – all, low and high relative humidity – and both regions, scaling rates are close to 2CC in the observations. In addition, rainfall intensities are higher for low relative humidity as compared to high relative humidity, typically increasing from 10 mm to 15-17 mm.

The CPMs tend to slightly overestimate the scaling rates for high relative humidity and the moister NL area. Conversely, they tend to underestimate scaling rates for low relative humidity, as visible for SFR-cent. Overall, however, scaling rates are rather accurate in the CPMs. There is a small overestimation of the intensity of rain; this could be partly related to an underestimation of rain frequency, as will be discussed below. But generally CPMs also have updrafts forced to occur at too large a scale (the km grid scale) due to insufficient turbulent mixing, causing also an overestimation of the rainfall intensity.


Compared to the CPMs, the results derived from the RCMs, are clearly worse. Scaling rates are on average accurate for high humidity, as visible for NL. But, as soon as the relative humidity drops, scaling rates start to become too low; this is most prominent for SFR-cent, where several RCMs display almost no dependency on dew point temperature. We note that the CPMs tend to struggle here too, with average dependencies close to the CC rate. But overall, the CPMs are clearly in better

agreement with observations and have lower inter-model spread.

As an example of where the CPMs improve on the RCMs, we plotted the difference in rainfall intensity between low and high relative humidity as a function of TD (Figure 8). In the observations, we obtained for NL and SFR-cent substantially more rainfall for low relative humidity in the TD range between 10 and 20 °C; peaking at around 15 °C, where hourly rainfall is 70-

100% more intense for low relative humidity. Most of the CPMs capture this intensification of precipitation for lower relative humidity, but they also tend to reach peak values at too low TD. In contrast, the RCMs do not consistently capture an intensification with lower relative humidity (with the exception again of HadRM3).


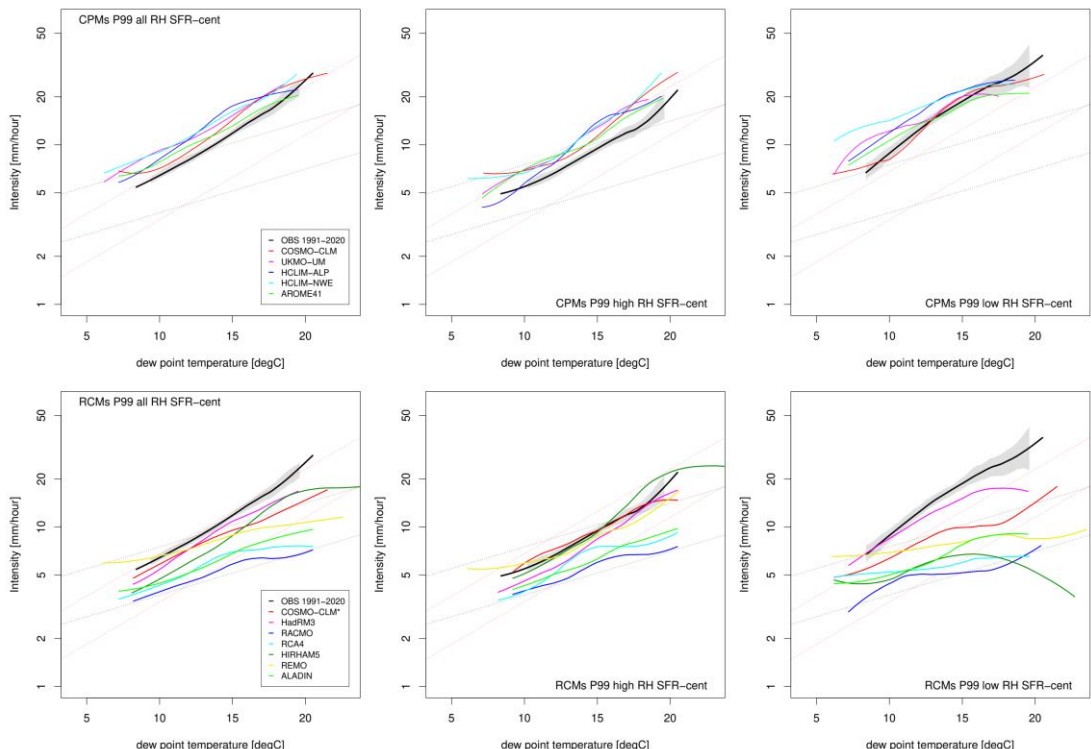

**Figure 6: Scaling of hourly precitation based on, from left to right, dew point temperature, dew point temperature for hours with low relative humidity, dew point temperature for hours with high relative humidity. Results are for the CPMs, and upper panels are for NL, and lower for SFR-cent. Lines filtered with LOESS filter (span = 0.5).**



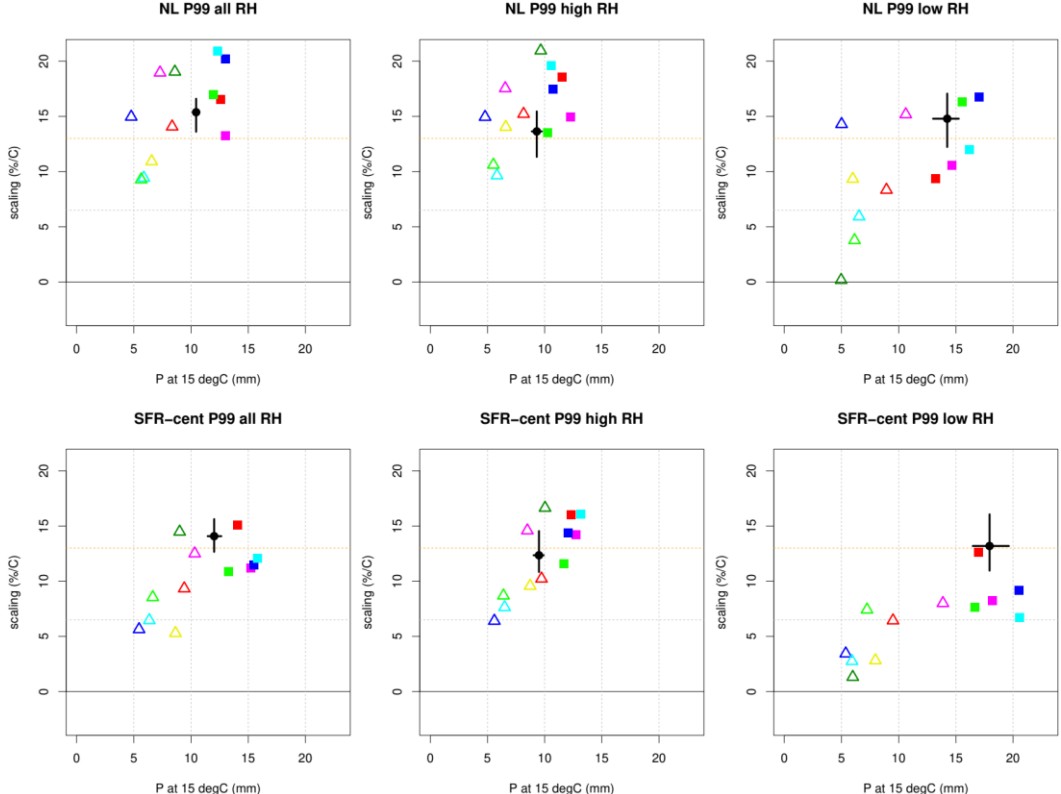

**Figure 7: Intensity at 15 °C versus scaling coefficient for the 99th percentile of hourly precipitation. Both are derived from a linear fit to log(P), fitted between 10 and 20 °C dew point. From left to right, NL, SFR-cent and SFR-med, and from top to bottom, all data, and data selected on low RH and high RH. CPM results are shown by solid squares, whereas RCM results are open triangles, and colouring is the same as Fig. 6.**

Besides rainfall intensity, which is conditional on the occurrence of rainfall, the frequency of rain is important as well. As expected, the frequency of rainfall is much higher for high relative humidities (of around 10-15%) than for low relative humidities (1-4%) (Figure 9, note the difference in scale). The wet hour frequency for SFR-cent and NL are surprisingly similar for high relative humidity, with values between 0.1 and 0.14 (Figure 9 and Fig. S12). This is not the case for low relative humidity, where the frequency of rain events is lower in SFR-cent than NL, possibly related to the lower relative humidity conditions in SFR-cent.





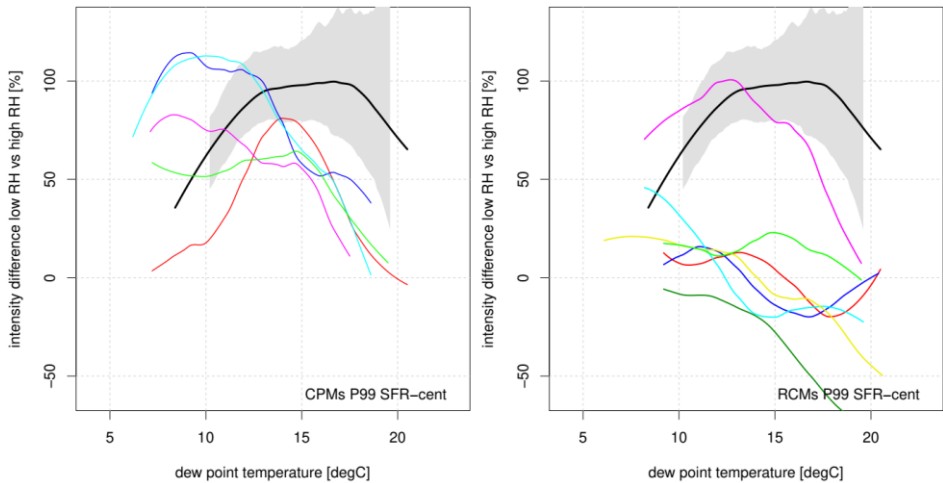


**Figure 8: Difference in intensity between low and high relative humidity (%) as a function of dew point temperature for SFR-cent (results for NL in the supplementary information). Colors are the same as Fig. 6.**

The CPMs generally reproduce the observed rainfall frequencies rather well (Figure 9). The CPMs tend to underestimate the

frequency of rainfall for high relative humidity, particularly for the high TD range (except for AROME). For low relative humidity they are approximately correct (except for COSMO in SFR-cent, which appears to underestimate). The RCMs clearly show more divergent behavior. In general, they strongly overestimate the frequency of rainfall (Figure 9)— a common deficiency of this type of model (Berthou et al., 2020; Ban et al., 2021; Lucas-Picher et al., 2021).

The dependence of the frequency of rainfall on relative humidity could have several causes, partly related to relative humidity directly affecting rain processes but also to confounding large-scale atmospheric conditions. Higher lifting condensation levels and higher values of convective inhibition associated with lower relative humidity could prevent an updraft from reaching its level of free convection and therefore suppress convective showers. However, by selecting on low relative humidity, large-scale atmospheric conditions are favoured that are more hostile to the development of convective showers. For instance, high

pressure systems cause low relative humidity as well as conditions in which showers are unlikely to develop, due to a lack of large-scale moisture convergence or stable tropospheric temperature profiles. The biases in model behaviour therefore cannot be easily attributed to the local cloud processes or to the confounding large-scale factors. Nevertheless, the overall negative bias in the simulation of the frequency of rainfall in the CPMs for high relative humidity, in particular for high TD, suggests that CPMs have difficulties in triggering convection in cases where the surface is relatively cold – noting that high relative

humidity implies a cold surface temperature for a given TD – and therefore for cases where the surface forcing is relatively weak.





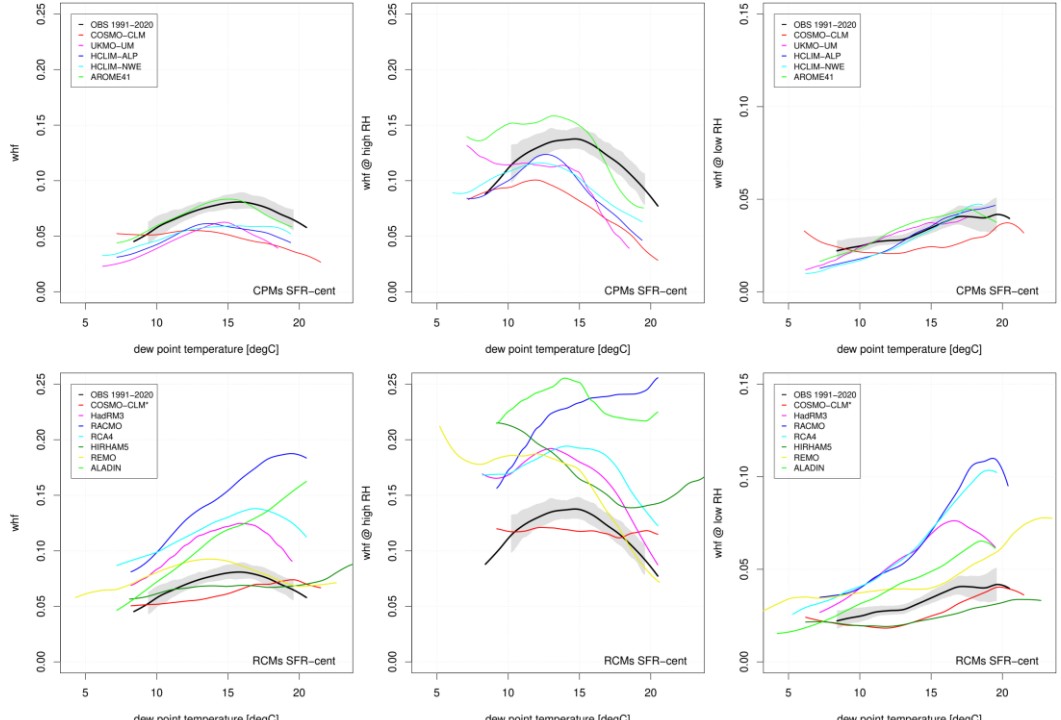

**Figure 9: Fraction of wet events as a function of dew point temperature, TD, for all hours (left), high relative humidity(middle) and low relative humidity RH (right) (note the difference in plotted range between low and high relative humidity). Lines filtered with LOESS filter (span = 0.3).**

The biases in the frequency of occurrence of rainfall may also affect results from the scaling analysis (Schär et al., 2016). We investigate this further in the discussion but note here that the main conclusions remain the same.

## 3.4 Area differences

Finally, we inspect differences between the extreme rainfall statistics in SFR-cent and NL. In addition to correctly representing humidity dependencies, a good representation of the systematic differences between these two contrasting regions adds confidence to the trustworthiness of these models.





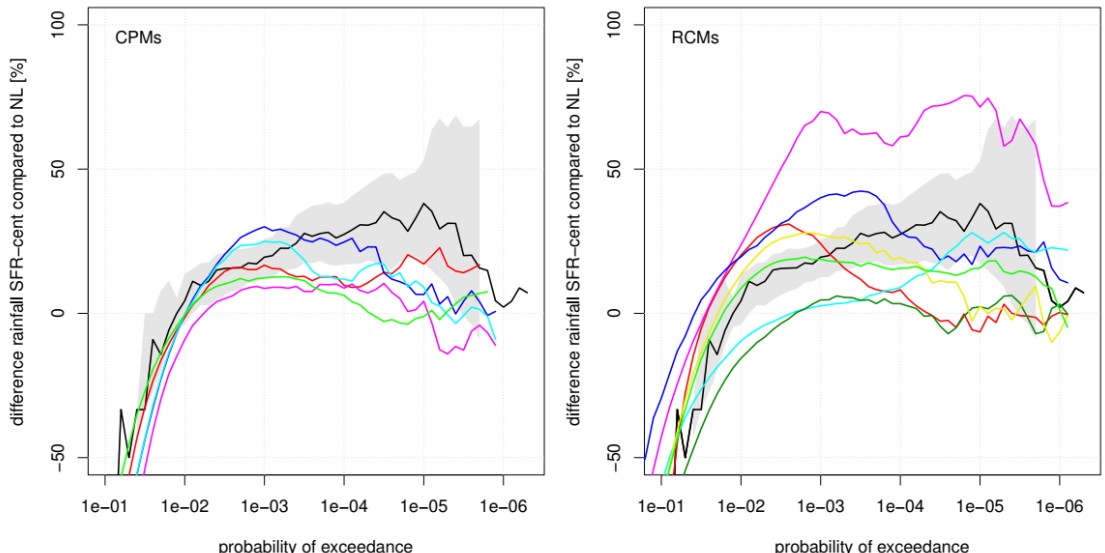

**Figure 10: Difference in hourly rainfall between SFR-cent and NI. The difference is the percentage difference (SFR-med minus NL) as function of the probability of exceedance estimated from pooling all stations within the region and for the summer half year.**

Before looking at the rainfall distribution differences between NL and SFR-cent, it is useful to inspect the differences in absolute and relative humidity in more detail. In general, dew point temperatures are slightly higher in SFR-cent as compared to NL; the climatological value is approximately 1 °C higher, and this difference decreases to almost zero for the most intense rain events (Figure 3, noting the difference in slope, and also Fig. S13). The differences in relative humidity are more pronounced. The median dew point depression is approximately 3-4 °C for the most extreme rainfall events in NL, whereas it

is 5-6 °C for SFR-cent with the largest difference for the most extreme rainfall. Thus, rainfall extremes in SFR-cent occur at slightly higher humidity and substantially lower relative humidity as compared to those in NL.

The difference in (extreme) rainfall is measured by $D_{SFR\text{-}cent-NL}$, which is defined as the fractional difference (in %) in hourly extreme rainfall (as a function of the probability of exceedance) with respect to the value for the NL. We note that results

beyond a probability of exceedance of ~$10^{-4}$ (~ 20 mm hour$^{-1}$) are less certain, as shown by the grey shaded areas based on the bootstrap procedure. Values for $D_{SFR\text{-}cent-NL}$ are positive for a probability of exceedance of less than $10^{-2}$, showing that rainfall extremes in central southern France produce 10-30% more rain in an hour than in the Netherlands. Based on the differences in humidity it appears likely that this is partly explained by the higher humidity values, and partly by the lower relative humidity. In particular, for the most extreme rainfall hours, the latter could be more important.


The CPMs capture the observed behaviour in $D_{SFR\text{-}cent-NL}$ well but are slightly too low for intermediate values of the probability of exceedance between $10^{-2}$ and $10^{-4}$, yet often within the uncertainty margins of the observations. For rarer events, with a





probability of exceedance less than ~$10^{-4}$, modelled $D_{\text{SFR-cent} - \text{NL}}$ values are (generally) 10-20% too low and are outside the uncertainty margins of the observations. We believe this to be primarily caused by the lower relative humidity in SFR-cent,
and the general tendency (small, but consistent) of the CPMs to underestimate rainfall extremes for low relative humidity.

The RCMs, on the other hand, display more diverse behaviour, and are generally further outside the uncertainty margins of the observations. However, the average across the RCM model ensemble does not appear to be worse than the CPM ensemble. This is at odds with the expectation that the too strong reduction of rainfall intensity to decreases in relative humidity should
lead to a more pronounced underestimation of $D_{\text{SFR-cent} - \text{NL}}$ as compared to the CPMs. We could not find a clear reason to explain this finding. It appears that $D_{\text{SFR-cent} - \text{NL}}$ is determined by a combination of different factors: the rain frequency and how it depends on relative humidity and other factors, the humidity dependencies of rain intensity, and the differences in underlying climatology. The interplay between these factors is complex, and most RCMs appear to suffer from compensating errors which are hard to disentangle.


As one example of this complexity, we discuss results from HIRHAM5. By all measures, HIRHAM5 has a very strong erroneous relative humidity dependency, only producing extreme rainfall for high relative humidity. Given the lower relative humidity of SFR-cent, one would therefore expect $D_{\text{SFR-cent} - \text{NL}}$ to be negative, but in fact values close to neutral are obtained. Part of this could be explained by the finding that, despite a mean negative bias, dew point temperatures are overestimated in
the upper range of the dew point temperature distribution by up to 2-4 °C (Fig. S2, and also visible in Figure 6 with scaling extending to much higher dew point temperatures). Similarly, the dew point depression is underestimated (Fig. S14). Thus it appears that compensating effects play a role here, and speculate that these may related to soil processes and soil memory, and how these interact with rainfall.

**4 Discussion**

Different measures have shown that the sensitivity of hourly extremes to relative humidity is essentially different in RCMs compared to CPMs, and that CPMs behave (much) more realistically in this respect. Lower relative humidity tends to lead to stronger rainfall extremes compared to high relative humidity: extremes co-occur on average with lower relative humidity (Figure 3) and at a given dew point temperature (given absolute humidity) rainfall intensities are higher for low relative
humidity (Figure 8).

It could be that these results are (partly) caused by a statistical effect caused by taking conditional percentiles. One may argue that low relative humidity suppresses showers when the large-scale conditions – for example, convergence at large scales – are not very favorable, taking light showers out of the rainfall distribution, but hardly affecting the occurrence of heavy





showers. In that case, the statistics of the extremes could still be the same, but the conditional percentiles would give an artificial increase (Schär et al., 2016). Also, this effect could play a role in explaining the differences between the models and the observations as the RCMs have large errors in the frequency of rain.

For this reason, we also considered the unconditional percentiles. The disadvantage of unconditional percentiles is that scaling
results become less robust, more dependent on the percentile, and more variable over the dew point temperature range (Figs. S5 and S15). But the difference is not very large, and is most pronounced for the lowest percentiles where the bootstrap sampling gives wider uncertainty estimates. The latter suggests that uncertainty in large-scale circulation conditions, which are captured by the resampling procedure, mostly affects the rain frequency leaving the conditional intensity distribution unaffected.


Taking the 99.9th percentile (roughly corresponding to the 99th percentile based on wet hours only) we still find that low relative humidity leads to stronger precipitation extremes. Yet, the degree of the effect is substantially reduced. In general, we now find that the extremes are 10-20% more intense in the case of low relative humidity (Figs. S15-17) – far below the 50-100% from estimated from the conditional percentiles. However, one should note that we now likely suffer from a reverse
compounding effect; low relative humidity is likely to co-occur with atmospheric high pressure systems. Thus, by comparing low with high relative humidity samples, we also sample the influence of large-scale circulation, for low relative humidity likely leading to a more hostile environment for convection to occur with less moisture convergence at larger scales and more stable lapse rates. Thus, we think that that this estimate is a lower bound and that in reality – comparing only systematic differences in near surface relative humidity, but otherwise similar other atmospheric drivers like stability and large-scale
convergence (Lenderink et al., 2017) – the sensitivity is in between the high estimate derived from the conditional rainfall distribution and the low estimate presented here.  This collaborates also with the finding that the most extreme events co-occur with lower relative humidity (Figure 3).

Physically, several processes could lead to an enhancement of rainfall extremes at lower relative humidity. At a fixed value of
the surface dew point temperature, lower relative humidity implies higher surface temperatures, so that low relative humidity is likely associated with atmospheric conditions with stronger heat forcing from the surface and higher values of convection inhibition (CIN). Likewise, lower relative humidity could be associated with mesoscale circulations transporting moisture to drier regions (Klein and Taylor, 2020; Hohenegger and Stevens, 2018). Lower relative humidity could lead to stronger cold pool dynamics because the deeper and drier (in terms of relative humidity) boundary layer causes more evaporation of rainfall
(Lochbihler et al., 2021).  The growth and collision of cold pools have been shown to play an important role in converging moisture and triggering new precipitation cells and this process could lead to bigger and more vigorous cloud systems





(Lochbihler *et al* 2021, Haerter and Schlemmer 2018). Also, it has been argued that deeper boundary layers lead to wider updrafts at the cloud base promoting stronger convection (Mulholland et al., 2021)

Conversely, low relative humidity could lead to weaker extreme rainfall as entrainment of dry air into the updraft motions could lead to strong cloud erosion and deep and dry boundary layers could lead to substantial evaporation of rain before it reaches the ground (Fowler *et al* 2021a, Derbyshire *et al* 2004). Also, the atmosphere may become too dry even to trigger moist convection if the lifting condensation level is not reached or values of CIN are too high. Since, in the limit of relative humidity approaching zero, there will be no moisture and therefore no rainfall, these processes that act to weaken extreme

rainfall must start to dominate at a certain (low) relative humidity value.

The causes of the relatively poor performance of the RCMs in representing the relative humidity dependencies are complex, but likely the consequence of the way the convection scheme interacts with its environment and lack of ability to capture local scale dynamics – with the parameterizations essentially treating this as a 1D problem whereas, since there is limited moisture

available in an atmospheric column, it is a 3D problem where moisture convergence is essential to sustain high rainfall rates. A number of processes leading to the enhancement of rain extremes with lower relative humidities are not parameterized or are poorly resolved in most RCMs, such as for instance the interactions between cold pools. Many mass flux based convection parameterizations use strong assumptions on the mass of air entering at cloud base, the lateral entrainment and detrainment of air into the cloud, lag memory from one time step to the next, and often suffer from numerical noise (Lenderink et al., 2004;

Hohenegger et al., 2009; de Rooy and Pier Siebesma, 2010; Yano et al., 2013). Although we cannot pinpoint a specific process being misrepresented in parameterizations or under-resolved at low resolution, it may be worthwhile using these relations to guide and evaluate further convection parameterization development, including upcoming schemes developed using machine learning techniques (Dwyer and O'Gorman, 2017; Gentine et al., 2018)

Further, we question whether differences in the humidity dependencies are reflected in the spatial differences between NL and SFR-cent. The climate in SFR-cent is most comparable to NL, with extreme rain occurring primarily in the summer months, but it is characterized by slightly higher absolute humidity (<1 °C higher TD), and considerably lower relative humidity (2-4 °C higher DPD). It is clear that CPMs show much more realistic humidity dependencies (by all metrics discussed here) with much smaller spread as compared to the RCMs; they are also more realistic and show less spread in simulating the difference

in extreme rain statistics between France and The Netherlands.

All CPMs underestimate the difference in extremes between SFR-cent compared to NL. This is likely related to the finding that they underestimate the sensitivity to humidity when the relative humidity is low. The dry bias in climatology – with too low relative humidity in most CPMs – could also play a role here. It may well be that low relative humidity is still enhancing



rainfall extremes for the Netherlands, while the lower relative humidity in SFR starts to limit the simulation of rainfall extremes
in the CPMs.

Finally, we note that several of the dependencies shown in this paper appear to be comparable in the two regions studied, the
Netherlands (NL) and central southern France (SFR-cent). Scaling based on dew point temperature shows ~2CC behaviour for

a large dew point temperature range. Dependencies on relative humidity are also broadly consistent with a general increase in
rainfall amount with decreasing relative humidity. The frequency of wet events is also similar between the regions, particularly
for high levels of relative humidity.

**5 Conclusions**

We evaluated 5 CPM and 7 RCM simulations – all driven by reanalysis data – using hourly rainfall observations over The

Netherlands and in southern France. We investigated extreme rainfall statistics as well as their sensitivities to absolute and
relative humidity.

We have shown that, on average for a model ensemble, CPMs have much better extreme hourly rainfall distributions in
comparison to RCMs, in agreement with earlier results. Importantly, CPMs more faithfully reproduce dependencies of hourly

rainfall extremes on absolute humidity (as measured by dew point temperature) and relative humidity (as measured by dew
point depression) than RCMs. RCMs show much greater spread, generally too low rainfall intensities, often (but not always)
underestimating the dependency on dew point temperature and are generally (except for HadRM3) unable to reproduce the
observed dependency of rainfall extremes on relative humidity.

The most striking result is the sensitivity of hourly rainfall to relative humidity, with all measures showing that lower relative
humidity leads to more intense rainfall for the climate conditions in NL and the inland part of southern France (SFR-cent).
This is particularly relevant in a climate change context as widespread decreases in relative humidity are expected in future
summers. The RCMs (with exception of one) cannot reproduce the observed extreme rainfall increase with decreasing relative
humidity and are therefore unreliable for extreme rainfall projections in a climate change setting. The CPMs qualitatively

reproduce the correct sensitivity to relative humidity, but still appear to underestimate the observed enhancement of rainfall
extremes with lower relative humidity. They are therefore more reliable, but may still underestimate climate change effects on
rainfall extremes.

In short, our results support the greater trustworthiness of CPM results as compared to RCMs in a climate change setting.

However, we also note deficiencies obtained in most CPMs: in general too low relative humidity, a negative bias in rain
frequency for high relative humidity with likely weak forcing from the surface, and biases in differences for the most extreme





hourly rainfall between The Netherlands and southern France. By comparing models with the observations using the metrics describing absolute and relative humidity dependencies as proposed in this paper, we can systematically improve climate models and obtain better future predictions of rainfall extremes.

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

*Author contributions.* GL wrote the paper and did most of the analysis. NB, EK and SB provided CPM data, and contributed to the text. EB and VEC analysed observational data for SFR. HJF and HdV contributed to the text and analysis.

825