# Peer review of "Are dependencies of extreme rainfall on humidity more reliable in convection-permitting climate models?"

_Hydrology and Earth System Sciences, 2024_

## Author Comment (AC2)

The manuscript by Lenderink et al presents a comprehensive evaluation of convection-permitting model (CPMs) performance for simulating rainfall (and extreme rainfall) relative to regional climate models (RCMs). The distributions of the rainfall, the rainfall-temperature relationship, and the rainfall-relative humidity relationship, and the differences between climatic regions are all better respected in CPMs as compared to RCMs. Moreover, the authors discuss in a very transparent manner the possible short comings of CPMs which will surely lead to improved model performance in the future as these short comings are addressed in future research.

As we move to a world where planning decisions are made on the basis of including more and more climate model results (rather than historical observations) building trust and confidence in the model results, as well as improving model results, is paramount. This manuscript addresses both these aims making it an important and impactful manuscript. This is a very thorough evaluation and hence I unreservedly look forward to this manuscript's publication.

I have comments below which I hope the authors will consider, but I should note, apart from errors I spotted in the supplementary figure numbers, the rest of the comments should be treated as suggestions. Although I feel it would be remiss of me to not suggest references that I am aware of, the authors should at no point feel obliged to cite them unless they feel they would enhance their manuscript. So again, I reiterate my comments should be treated as suggestions, not prescriptions.

Thank you very much for your comments. Below, I will briefly answer to your questions and comments in red.

General comments:

The robustness of the super Clausius-Clapeyron scaling wasn't explicitly stated in the abstract or conclusions and this could be added if the authors wish.

We will add a few lines on this in the conclusions.

The interesting result of rainfall being more intense at lower RH appears is well discussed. I had a thought that this could also be affected by the drop in temperature across events which becomes greater with rarer events (Figure 4 in Barbero et al) but I assume that picking the temperature before the rainfall event negates this effect? The authors may wish to comment on this in the manuscript but shouldn't feel obliged to. Apologies if this was already mentioned and I missed it.

That's an interesting thought, and thank you for pointing out Figure 4 in that paper. Indeed, more intense events are associated with stronger temperature drops. There could be several causes of that. One is that stronger temperature drops are associated with stronger large-scale disturbances, perhaps with colder higher level temperature (and therefore higher CAPE). The other is that stronger temperature drops could be well related to stronger cold pools, and since these are (partly) driven by evaporation of rain boundary layer relative humidity comes into play. In fact even the temperature drop is a rather good measure to distinguish events satisfy 2CC relation versus lower scaling events. Below, you will find scaling derived from sub-selecting on the temperature drop between 2 hours before the rainfall event and one hour after. It is clear that larger temperature drops are associated with stronger rainfall extremes. But also sub-selecting on negative temperature drops most of the super CC behavior disappears. We will add this plot to the supplement.

[Figure]

*Figure 1. Scaling on sub-selections of data characterized by the temperature drop between two hour before and one hour after the rainfall measurement. From left to right, a temperature rise (negative drop), temperature drop, and temperature drop exceeding 2 degrees.*

Barbero, R., Westra, S., Lenderink, G., Fowler, H.J., 2017. Temperature-extreme precipitation scaling: a two-way causality? Int. J. Climatol. 38, e1274–e1279. https://doi.org/10.1002/joc.5370

Line by line comments:

Line 45: The reference below showed RCM parameterisations fail to capture spatial dependencies of rainfall.

Li, J., Wasko, C., Johnson, F., Evans, J.P., Sharma, A., 2018. Can Regional Climate Modeling Capture the Observed Changes in Spatial Organization of Extreme Storms at Higher Temperatures? Geophys. Res. Lett. 45, 4475–4484. https://doi.org/10.1029/2018GL077716

Thank you for this reference, which will be added.

Line 52: Where you say "when future changes are dominated by simple thermodynamics" when is this exactly? I think it is for long duration (daily) rare rainfalls, but it would be nice to state this explicitly. Having said that, if a short duration event is embedded in a long duration event, then convection permitting modelling remains crucial.

This is indeed the case. But we agree that the distinction between the two situations is not so clear. We can add there if two conditions are met: "large-scale dynamics control the event and these large-scale dynamics do systematically change" .  But to be fair, we think that is more a statistical statement, and it will be hard to proof this to be the case for an event.

Line 72: "depend on the temperature measure used" – I wonder if something could also be said for the need to make sure the temperature is the one driving the event by making sure rainfall events are independent when captured. When this is done this gives reduced variability in the rainfall-temperature scaling. This conclusion in Visser et al is ultimately the outcome of these three manuscripts.

Barbero, R., Westra, S., Lenderink, G., Fowler, H.J., 2017. Temperature-extreme precipitation scaling: a two-way causality? Int. J. Climatol. 38, e1274–e1279. https://doi.org/10.1002/joc.5370

Schleiss, M., 2018. How intermittency affects the rate at which rainfall extremes respond to changes in temperature. Earth Syst. Dyn. 9, 955–968. https://doi.org/10.5194/esd-9-955-2018

Visser, J.B., Wasko, C., Sharma, A., Nathan, R., 2021. Eliminating the "hook" in Precipitation-Temperature Scaling. J. Clim. 34, 9535–9549. https://doi.org/10.1175/JCLI-D-21-0292.1

We indeed made no effort to distinguish between events. We will mention that in the discussion. We also note that in this paper Lenderink G, Barbero R, Loriaux JM, Fowler HJ (2017) https://doi.org/10.1175/JCLI-D-16-0808.1 we clustered rainfall into events, and found similar scaling.

Line 95: it might be worth mentioning that such reductions in relative humidity have also been observed and not just modelled?

Denson, E., Wasko, C., Peel, M.C., 2021. Decreases in relative humidity across Australia. Environ. Res. Lett. 16, 074023. https://doi.org/10.1088/1748-9326/ac0aca

This is definitely true, and we will mention that. Unfortunately, we are unaware of such a paper for Europe.

Line 170: I could be wrong here, but was Pmean used at all in the manuscript? I got the impression that only Psample was used so I was a bit confused to why these were both defined.

Indeed, the paper uses the sample value. But we did all analysis also with the mean value. We will add a figure to supplement to show the difference between sample and mean value.

Line 226: Is the "all time-steps" including zero rainfall time-steps or just the wet time steps?

This is including dry hours; we will add that.

Section 3.1: There was a really comprehensive discussion of why some models were better/worse for the RCMs but for the outlier for the CPMs (UKMO-UM), it wasn't discussed as to the reason why this might be the case – this could be added around lines 235-240.

If this is known by the Met Office I will add this, but due to holiday time no information yet from them.

Figure 3 caption: The parentheses for the last line are probably not required.

OK

Line 251: The higher intermittency I assume may lead to drier soils and less humidity? The link to the above sentence that discuss model performance was missing because the above sentences talk about the model performance in terms of simulating temperature.

Lines 252-255: I wasn't really sure what figure I was supposed to be looking at, so I didn't really understand this part I am sorry. I think Figure 3 was introduced but then the discussion shifted to discussing Figure S2 and this wasn't clear.

We will clarify this paragraph, better explaining that we look at humidity (dew point temperature as dew point depression), and the link to soil moisture. We will also add the reference to the appropriate Figure to the text.

Lines 266-274: I am not sure the results of the DPD for median and 80[th] percentile (middle and right columns) were contrasted in the text?

We will add a sentence that it is robust, so both median and 80[th] percentile showing very similar results

Line 319: "similar to a quantile difference plot" - I think an "a" is missing in this sentence.

OK

Line 302: Again, the link of soil memory and runoff to the temperature simulated is missing.

I am not sure whether I understand this because we do not consider temperature as a scaling parameter. But will clarify this in earlier lines (252-255).

Line 348: Can the figure numbers in Supplementary be listed? I think this sentence needs rewording because when you say "plots for individual models are presented in supplemtnary" this implies that Figure 5 is pooled model results, but they are just observations correct? Maybe saying "plots comparing observations to individual models are presented in supplementary and are summarised in Figure 6?" would be clearer.

Indeed, Figure5 is observations only. We will reformulate this.

Line 349: "For further investigation…" should this say, "For further investigation and comparison to the model simulations"? I think it could be made explicit that you are now shifting from observations to also looking at model output.

Yes, indeed. Thank you

Figure 6: I think the caption is wrong – it says NL is presented, but the text at Line 353 says NL is in the supplement. The caption should also mention the percentile analysed. Finally, I think you mean Figure S10 and not Figure S6?

Thanks for spotting this error. We will correct this error.

Line 350: Although the split of low and high RH was explicitly stated in the methods it could be mentioned here again to remind the reader?

A good suggestion

Line 417 – I think you should mention the figure number in supplementary.

OK

Line 446 – "Area" could maybe be "regional" differences?

A good suggestion

---

## Author Response (AR1)

Dear Editor and reviewers.

I would like to thank you for all your contributions, and useful feedback on the manuscript. I am really happy with the revisions.

Apart from the comment by the reviewers, I got the request to update the figures to more colorblind friendly colors. After consulting the editor office, we decided to use color proposed by Wong (https://davidmathlogic.com/colorblind/#%23D81B60-%231E88E5-%23FFC107-%23004D40). This implies that all figures, except Figure 5, have been remade. In doing so, we also spotted a (very) small error in Figure 10 (which including results from April in the previous version, but differences are very minor and do not affect the conclusions.). In Figure 5, we added the percentiles to the lines (in left panel), and a better description of the different lines.

Below we give a detailed response to reviewers comments, with our response marked in red.

Best regards

Geert Lenderink

**Reviewer A**

The manuscript by Lenderink et al presents a comprehensive evaluation of convection-permitting model (CPMs) performance for simulating rainfall (and extreme rainfall) relative to regional climate models (RCMs). The distributions of the rainfall, the rainfall-temperature relationship, and the rainfall-relative humidity relationship, and the differences between climatic regions are all better respected in CPMs as compared to RCMs. Moreover, the authors discuss in a very transparent manner the possible short comings of CPMs which will surely lead to improved model performance in the future as these short comings are addressed in future research.

As we move to a world where planning decisions are made on the basis of including more and more climate model results (rather than historical observations) building trust and confidence in the model results, as well as improving model results, is paramount. This manuscript addresses both these aims making it an important and impactful manuscript. This is a very thorough evaluation and hence I unreservedly look forward to this manuscript's publication.

I have comments below which I hope the authors will consider, but I should note, apart from errors I spotted in the supplementary figure numbers, the rest of the comments should be treated as suggestions. Although I feel it would be remiss of me to not suggest references that I am aware of, the authors should at no point feel obliged to cite them unless they feel they would enhance their manuscript. So again, I reiterate my comments should be treated as suggestions, not prescriptions.

Thank you very much for your comments. Below, I will briefly answer to your questions and comments in red.

General comments:

The robustness of the super Clausius-Clapeyron scaling wasn't explicitly stated in the abstract or conclusions and this could be added if the authors wish.

We added this to the first paragraph of the conclusions, but since it was already mentioned at the end of the discussion we removed it there. Instead, we added in the discussion a paragraph on the relation with temperature drop as suggested below.

The interesting result of rainfall being more intense at lower RH appears is well discussed. I had a thought that this could also be affected by the drop in temperature across events which becomes greater with rarer events (Figure 4 in Barbero et al) but I assume that picking the temperature before the rainfall event negates this effect? The authors may wish to comment on this in the manuscript but shouldn't feel obliged to. Apologies if this was already mentioned and I missed it.

That's an interesting thought, and thank you for pointing out Figure 4 in that paper. Indeed, more intense events are associated with stronger temperature drops. There could be several causes of that. One is that stronger temperature drops are associated with stronger large-scale disturbances, perhaps with colder higher level temperature (and therefore higher CAPE). The other is that stronger temperature drops could be well related to stronger cold pools, and since these are (partly) driven by evaporation of rain boundary layer relative humidity comes into play. In fact even the temperature drop is a rather good measure to distinguish events satisfy 2CC relation versus lower scaling events. Below, you will find scaling derived from sub-selecting on the temperature drop between 2 hours before the rainfall event and one hour after. It is clear that larger temperature drops are associated with stronger rainfall extremes. But also sub-selecting on negative temperature drops most of the super CC behavior disappears. We will add this plot to the supplement (Fig. S18).

[Figure]

*Figure 1. Scaling on sub-selections of data characterized by the temperature drop between two hour before and one hour after the rainfall measurement. From left to right, a temperature rise (negative drop), temperature drop, and temperature drop exceeding 2 degrees.*

Barbero, R., Westra, S., Lenderink, G., Fowler, H.J., 2017. Temperature-extreme precipitation scaling: a two-way causality? Int. J. Climatol. 38, e1274–e1279. https://doi.org/10.1002/joc.5370

We added a small paragraph at the end of the discussion.

Line by line comments:

Line 45: The reference below showed RCM parameterisations fail to capture spatial dependencies of rainfall.

Li, J., Wasko, C., Johnson, F., Evans, J.P., Sharma, A., 2018. Can Regional Climate Modeling Capture the Observed Changes in Spatial Organization of Extreme Storms at Higher Temperatures? Geophys. Res. Lett. 45, 4475–4484. https://doi.org/10.1029/2018GL077716

Thank you for this reference, which is added in the introduction.

Line 52: Where you say "when future changes are dominated by simple thermodynamics" when is this exactly? I think it is for long duration (daily) rare rainfalls, but it would be nice to state this explicitly. Having said that, if a short duration event is embedded in a long duration event, then convection permitting modelling remains crucial.

This is indeed the case. But we agree that the distinction between the two situations is not so clear. We reformulated to " Indeed, when future changes are dominated by simple thermodynamic factors (when increases from moisture dominate dynamical changes or when dynamical changes occur at scales well resolved by the RCMs) then CPMs often project similar changes to RCMs". But to be fair, we think that is more a statistical statement, and it will be hard to proof this to be the case for an event.

Line 72: "depend on the temperature measure used" – I wonder if something could also be said for the need to make sure the temperature is the one driving the event by making sure rainfall events are independent when captured. When this is done this gives reduced variability in the rainfall-temperature scaling. This conclusion in Visser et al is ultimately the outcome of these three manuscripts.

Barbero, R., Westra, S., Lenderink, G., Fowler, H.J., 2017. Temperature-extreme precipitation scaling: a two-way causality? Int. J. Climatol. 38, e1274–e1279. https://doi.org/10.1002/joc.5370

Schleiss, M., 2018. How intermittency affects the rate at which rainfall extremes respond to changes in temperature. Earth Syst. Dyn. 9, 955–968. https://doi.org/10.5194/esd-9-955-2018

Visser, J.B., Wasko, C., Sharma, A., Nathan, R., 2021. Eliminating the "hook" in Precipitation-Temperature Scaling. J. Clim. 34, 9535–9549. https://doi.org/10.1175/JCLI-D-21-0292.1

We indeed made no effort to distinguish between events. We will mention that in the discussion. We also note that in this paper Lenderink G, Barbero R, Loriaux JM, Fowler HJ (2017) https://doi.org/10.1175/JCLI-D-16-0808.1 we clustered rainfall into events, and found similar scaling. We added in the text (last paragraph discussion): In this study, we did not separate out hourly rainfall data into rainfall events (such as in Lenderink et al. 2017; Visser et al. 2021). Recent research showed that an event based analysis could increase the robustness of scaling (Visser et al. 2021). For instance, they found that the typical "hook shape" from positive to negative scaling rates at high temperatures (and low relative humidity) found in many studies can be attributed to a reduction of the rainfall event duration, while event peak intensities still increase with temperature. Thus, it would be interesting to repeat this analysis based on rainfall events.

Line 95: it might be worth mentioning that such reductions in relative humidity have also been observed and not just modelled?

Denson, E., Wasko, C., Peel, M.C., 2021. Decreases in relative humidity across Australia. Environ. Res. Lett. 16, 074023. https://doi.org/10.1088/1748-9326/ac0aca

This is definitely true, since we are studying Europe this reference appears not so well suited. But we included to recent studies looking at a global perspective "Even more so, decreases in relative humidity are already widely observed over the land even outside predicted model changes (Vicente-Serrano et al. 2018; Simpson et al. 2024). "

Line 170: I could be wrong here, but was Pmean used at all in the manuscript? I got the impression that only Psample was used so I was a bit confused to why these were both defined.

Indeed, the paper uses the sample value. But we did the analysis also with the mean value. We added 2 figures to supplement to show the difference between sample and mean value (Fig S19-20).

Line 226: Is the "all time-steps" including zero rainfall time-steps or just the wet time steps?

This is including dry hours; we added this

Section 3.1: There was a really comprehensive discussion of why some models were better/worse for the RCMs but for the outlier for the CPMs (UKMO-UM), it wasn't discussed as to the reason why this might be the case – this could be added around lines 235-240.

According to EK, coauthor of this paper, UKMO does not use a shallow convection scheme, which may results in overcompensation by resolved convection. Added this.

Figure 3 caption: The parentheses for the last line are probably not required.

OK

Line 251: The higher intermittency I assume may lead to drier soils and less humidity? The link to the above sentence that discuss model performance was missing because the above sentences talk about the model performance in terms of simulating temperature.

Lines 252-255: I wasn't really sure what figure I was supposed to be looking at, so I didn't really understand this part I am sorry. I think Figure 3 was introduced but then the discussion shifted to discussing Figure S2 and this wasn't clear.

Above two comments: We added a sentence that too dry soils could lead to too low dew points and too high dew point depressions, and a reference to figure 3 at the appropriate place.

Lines 266-274: I am not sure the results of the DPD for median and 80[th] percentile (middle and right columns) were contrasted in the text?

We added "Both median and 80[th] percentile of the distribution of DPD show similar results (middle and right panels)."

Line 319: "similar to a quantile difference plot" - I think an "a" is missing in this sentence.

OK

Line 302: Again, the link of soil memory and runoff to the temperature simulated is missing.

This has been clarified before.

Line 348: Can the figure numbers in Supplementary be listed? I think this sentence needs rewording because when you say "plots for individual models are presented in supplemtnary" this implies that Figure 5 is pooled model results, but they are just observations correct? Maybe

saying "plots comparing observations to individual models are presented in supplementary and are summarised in Figure 6?" would be clearer.

Indeed, Figure5 is observations only. We made this more clear

Line 349: "For further investigation…" should this say, "For further investigation and comparison to the model simulations"? I think it could be made explicit that you are now shifting from observations to also looking at model output.

Yes, indeed. Thank you, changed

Figure 6: I think the caption is wrong – it says NL is presented, but the text at Line 353 says NL is in the supplement. The caption should also mention the percentile analysed. Finally, I think you mean Figure S10 and not Figure S6?

Thanks for spotting this error. We corrected this

Line 350: Although the split of low and high RH was explicitly stated in the methods it could be mentioned here again to remind the reader?

Added a reference to the methods.

Line 417 – I think you should mention the figure number in supplementary.

OK, added Fig. S11.

Line 446 – "Area" could maybe be "regional" differences?

Changed this; thank you.

**Reviewer B**

This manuscript provides a comprehensive analysis and evaluation of ensembles of RCMs with parameterized convection and CPMs with explicitly resolved convection for two regions that are likely to see changes in rainfall convective rainfall extremes. The topic is well researched with a comprehensive background section, the analysis is reasonable, and the results are convincing. Further, the authors provide an insightful discussion on their results and the paper is almost entirely devoid of grammar issues. I think the paper is valuable and an interesting contribution to the literature and, accordingly, I recommend the paper be published pending the resolution of a handful of minor comments.

Thank you very much for your review, and helpful comments. Below you will find a response to them.

I have two key questions that the authors should address so as to better contextualize their results and conclusions.

1. What is the specific resolution (or range of resolutions) of the CPMs used in the study? The RCM resolution is listed at 12km, but I don't recall seeing the CPM resolution described in either the manuscript or the supplementary material. This is important as numerous studies show that for convection, diurnal convection in particular, 2km resolution is generally superior at reproducing the timing and intensity of diurnal convection (e.g., https://agupubs.onlinelibrary.wiley.com/doi/full/10.1002/2014RG000475). If the models used in this study are coarser resolution than 2km, I think that's fine since the main consensus within the community is at 4km cutoff. That said, some context here would really improve the manuscript. On a similar note, any comments the authors can provide on the LSMs used in the models described in study would be helpful since at sub 4km resolutions, simulated convection becomes increasingly tied to the LSM. (e.g., are there any known biases in the LSMs that might impact the results or explain some of the shortcomings in the CPMs?).

   The resolution of the CPMs model is 2.2 km (COSMO ) and 2.5 km (UKMO, HCLIM and AROME). This was indeed missing. We added in the paper "The resolution of the models is 2.2 km for COSMO and 2.5 for the other models. With this resolution, the CPMs are expected to have substantially improved (non-hydrostatic) mesoscale dynamics compared RCMs (see discussion in Prein et al. 2015)."

   We do agree that the soil scheme has a large influence on rainfall, for instance because land surface heterogeneity may influence the triggering on rain systems. But at the same time LSMs are so complex that it is hard to pinpoints at problems. It may be the formulation of the LSM, but also the underlying data base of soil and vegetation properties can strongly affect results. We therefore think this is outside the scope of the paper. But we added in the discussion "Here, we also emphasize the importance of the land surface scheme and how it interacts with convective processes; for instance, how convection reacts to surface heterogeneities and associated mesoscale circulations, and how rainfall is divided into soil water storage and runoff (Prein et al. 2015; Halladay et al. 2024)."

2. What is the generalized modality of the convection that makes up the extreme rainfall events (95 percentile, e.g.) in the regions described in the study? Diurnal air-mass "pop-corn" ordinary cells? Organized MCSs? Terrain-initiated convection? Convection forced by or embedded within synoptic scale systems (e.g., fronts)? Mix of all of the above? The authors provide an excellent discussion on the complexities of convection and extreme rainfall and how changes in absolute and relative humidity in the surface can influence or be related to cloud-scale dynamics, however without information on what convective modes are being simulated, this discussion is without context. For example, the authors discuss convective plume size vs. dry air entrainment as a physical reason for the observed behavior of extreme rain vs. dew point depression. However, my understanding is that this process is mostly only relevant for continental convection associated with very dry mid-level air where wider convective plumes associated with larger surface DPDs can protect the convective core from dry-air entrainment in the mid-levels. If, for instance, the convection most associated with extreme rain in the NL or SFR was more characteristic of air-mass convection with very moist conditions throughout the atmospheric column, then this process likely isn't a great explanation for the DPD/extreme rain relationships observed. To be clear, I am not arguing that this process *isn't* relevant in this instance, I'm just arguing that without added context on storm modality, the authors' speculation on physical processes carries less weight. A brief discussion of predominant storm modality, and perhaps an example figure showing a snapshot of simulated rain-rate compared to radar image for an extreme rain event would be extremely helpful to the reader and would improve the manuscript. Such a discussion would also relate back to CPM resolution, since the differences between 4km and 2km resolutions would be quite different for a synoptically forced MCS vs air-mass thunderstorms.

These are good questions, and we thank the reviewer for asking them. Definitely, the statistics we get are based on many different events. We tried to rule out large orographic effect by focusing on stations with altitude below 400 m. For The Netherlands orography definitely does not play a large role as most of the country is very flat. In a previous paper we looked at the large-scale conditions associated with the events (Lenderink G, Barbero R, Loriaux JM, Fowler HJ (2017). https://doi.org/10.1175/JCLI-D-16-0808.1). Relative humidity associated with extreme events is shown in Figure 6 of that paper. We found that the extreme statistics are dominated by larger scale events, where rainfall is embedded into large scale disturbances, with substantial large-scale lifting (e.g. Figs 6 and 8 of the paper). We also found that rainfall itself is not more intense when it occurs in larger clusters – the hourly rainfall distribution is exactly the same for small and large clusters – but since rainfall occurs in much larger areas they dominate the extreme statistics (Fig 3cd). This also appears to collaborate with the finding from Large-Eddy Simulation that large-scale lifting mostly affects the size of the systems, but that instability mostly affects intensity (Loriaux JM, Lenderink G, Siebesma AP (2017) https://doi.org/10.1175/JCLI-D-16-0381.1).

We added in the paper at the end of the methods section "An earlier analysis for the data from the Netherlands showed that the far majority of extreme rainfall measurements are connected to relatively large-scale events, where rainfall occurs at many neighboring stations (Lenderink et al. 2017). Yet, small-scale events appear to have similar statistical distributions (see Figure 3 of that paper). It is the fact that there are many more hours

with rainfall in large rainfall events that causes the domination of the type of convective events in the overall extreme statistics. In addition, most rainfall extremes occur in conditions with substantial large-scale rising motions indicative of substantial synoptic scale forcings."

Finally, we note scaling is rather robust, and that sub-selecting on large-scale circulation types only moderately effects the results (see Figure below). Thus, it appears that many of the sensitivities we find are not so dependent on the type of convection and associated large-scale circulation patterns. Yet, we definitely agree with the reviewer that it would be interesting to investigate this further, and see how storm type is influencing the response to warming.

[Figure]

*Figure 2. Scaling of hourly rainfall extremes, sub-selected on circulation type. From left to right, westerly flow, easterly flow, northly flow and southerly flow patterns over the Netherlands.*

**Minor comments:**

1. Why not look directly at CAPE or CIN? When discussing /speculating on some of the convective processes and how they relate to changes in absolute and relative humidity? Were these data simply not available?
   No, this data is not available. Some models may have it, but there are definitely no reliable soundings for all stations, so we have to rely on reanalysis data. In earlier work we looked at CAPE and found that is quite sensitive to the quality of the surface observations and the time lag with the precipitation (Loriaux JM, Lenderink G, Siebesma AP (2016) https://doi.org/10.1002/2015JD024274; e.g. Figure 7). So, we think this is a much less robust measure.

2. Why use DPD instead of RH directly?
   This could have been done, but we prefer to keep everything in a "temperature" space. E.g. from the DPD it is easy to derive an approximate value for the cloud base given an undiluted surface parcel (LCL ~DPD*100 m).

3. I'm unclear as to the use of the 5x5 pixel sampling, and how it was used in the data comparisons, the mean was computed, but I don't know if it was actually used?

This point was also noted by the other reviewer. In addition to the "sample" based value, we also did the analysis on the mean, to investigate whether dependencies on spatial average could affect our results, and explain part of the differences between the CPMs and the RCMs. But, it turned out that differences are rather marginal in most plots. We added two figures in the supplement (Fig S19-S20).

4. Line 246/247: dew-point temperature, TD, and TDD should be modified: TD and TDD. (no need to repeat dew-point temperature).
Done

5. It is my understanding that the analysis performed comes from simulations that the authors themselves did not perform, but this point is not entirely clear as it relates to the CPMs. I suggest the authors clarify this point.
We made this more clear in the acknowledgements.